# Triamterene Functions as an Effective Nonsense Suppression Agent for MPS I-H (Hurler Syndrome)

**DOI:** 10.3390/ijms24054521

**Published:** 2023-02-24

**Authors:** Amna Siddiqui, Halil Dundar, Jyoti Sharma, Aneta Kaczmarczyk, Josh Echols, Yanying Dai, Chuanxi Richard Sun, Ming Du, Zhong Liu, Rui Zhao, Tim Wood, Shalisa Sanders, Lynn Rasmussen, James Robert Bostwick, Corinne Augelli-Szafran, Mark Suto, Steven M. Rowe, David M. Bedwell, Kim M. Keeling

**Affiliations:** 1Department of Biochemistry & Molecular Genetics, Heersink School of Medicine, University of Alabama at Birmingham, Birmingham, AL 35294, USA; 2Next Generation Sequencing Transplant Diagnostics, Thermo-Fisher Scientific, West Hills, CA 91304, USA; 3Cystic Fibrosis Research Center, Heersink School of Medicine, University of Alabama at Birmingham, Birmingham, AL 35294, USA; 4Division of Infectious Diseases, Heersink School of Medicine, University of Alabama at Birmingham, Birmingham, AL 35294, USA; 5ARUP Laboratories, Department of Pathology, University of Utah, Salt Lake City, UT 84112, USA; 6Greenwood Genetic Center, Greenwood, SC 29646, USA; 7Department of Pediatrics, University of Colorado Anschutz Medical Campus, Aurora, CO 80045, USA; 8Southern Research, Birmingham, AL 35205, USA

**Keywords:** triamterene, MPS I-H, nonsense suppression, readthrough, *IDUA*-W402X

## Abstract

Mucopolysaccharidosis I-Hurler (MPS I-H) is caused by the loss of α-L-iduronidase, a lysosomal enzyme that degrades glycosaminoglycans. Current therapies cannot treat many MPS I-H manifestations. In this study, triamterene, an FDA-approved, antihypertensive diuretic, was found to suppress translation termination at a nonsense mutation associated with MPS I-H. Triamterene rescued enough α-L-iduronidase function to normalize glycosaminoglycan storage in cell and animal models. This new function of triamterene operates through premature termination codon (PTC) dependent mechanisms that are unaffected by epithelial sodium channel activity, the target of triamterene’s diuretic function. Triamterene represents a potential non-invasive treatment for MPS I-H patients carrying a PTC.

## 1. Introduction

A nonsense mutation generates an in-frame premature termination codon (PTC) within the open reading frame of an mRNA. A PTC reduces the amount of functional protein translated from an mRNA in two ways. First, a PTC terminates translation before a full-length protein can be generated, resulting in a truncated polypeptide that lacks normal function or is unstable. Second, a PTC can activate nonsense-mediated mRNA decay (NMD), a conserved eukaryotic cellular pathway that targets PTC-containing mRNAs for degradation, reducing the pool of mRNA available for translation. Significantly, 11% of all gene lesions associated with a genetic disease generate an in-frame PTC [1], indicating that millions of patients carry this type of mutation.

One way to overcome the effects of nonsense mutations on gene expression is to suppress translation termination at PTCs, allowing partial levels of full-length, functional protein to be restored. PTC suppression occurs when an aminoacyl-tRNA that base-pairs with two of the three nucleotides of a PTC, known as a near-cognate aminoacyl-tRNA, becomes accommodated into the ribosomal acceptor site and its amino acid is added to the nascent polypeptide at the PTC position [2,3]. This mechanism, often termed “readthrough”, permits translation elongation to continue downstream of the PTC in the correct ribosomal reading frame, generating a full-length polypeptide. The use of small molecules that promote the readthrough of PTCs is currently being explored as a precision/personalized medicine approach to restore the deficient protein in patients who harbor a nonsense mutation. However, the readthrough agents that are currently available for clinical use are unlikely to be effective enough to treat most genetic diseases. Factors that can limit the effectiveness of current readthrough agents include: the mRNA context surrounding a PTC [4,5], the nature of the amino acid that becomes incorporated at a PTC during readthrough [2,3], the efficiency of NMD that limits mRNA availability for readthrough [6,7,8], and the level of protein function required to alleviate a disease phenotype. More effective and safe readthrough agents are needed that can overcome these limitations and can be applied to diseases for which rescue of partial levels of protein function is sufficient to normalize the phenotype.

One such disease that could benefit from readthrough therapies is Mucopolysaccharidosis I (MPS I). MPS I is caused by a deficiency of the lysosomal enzyme, α-L-iduronidase (encoded by the *IDUA* gene), that participates in the catabolism of the glycosaminoglycans (GAGs) dermatan sulfate and heparan sulfate. α-L-iduronidase insufficiency results in the cumulative storage of these partially digested GAGs in lysosomes. Complete loss of α-L-iduronidase leads to the onset of the severe, Hurler form of MPS I (MPS I-H), which is characterized by progressive development of abnormalities in the cartilage, bone, heart, spleen, liver, lung, and neurological tissues, ultimately resulting in a reduced lifespan. MPS I-H represents an excellent candidate for readthrough therapies since the majority of MPS I-H patients carry a nonsense mutation [9] and even low levels of normal α-L-iduronidase activity (<1%) can significantly attenuate MPS I-H phenotypes [10].

To identify more effective readthrough compounds that could potentially be used to treat MPS I-H (and other diseases), we performed a screen of ~10,000 low-molecular weight compounds for those that suppress PTCs. This screen was performed using NanoLuc reporter cell lines that were previously shown to be sensitive to changes in both readthrough and mRNA abundance, and used to identify eRF1 degrader molecules, a new class of readthrough agent [11]. Our current screen was performed with each compound alone and in the presence of a low dose of the aminoglycoside, G418. Although G418 is a potent readthrough compound, it cannot be used clinically due to its toxicity. However, it was previously shown that G418 can be used to identify enhancer molecules that do not mediate readthrough themselves but enhance the activity of other readthrough compounds [12]. In addition, G418, which induces readthrough by increasing ribosomal misreading, has been shown to synergistically increase readthrough when combined with compounds that mediate readthrough via other mechanisms (such as eRF1 [11] and eRF3 [13] degraders). We subsequently examined primary hits from this screen for the ability to suppress nonsense mutations associated with MPS I-H.

Triamterene, an FDA-approved, potassium-sparing, diuretic used to treat hypertension, was identified as a hit from our screen. Triamterene increased α-L-iduronidase activity in a dose-dependent manner and correspondingly decreased GAG storage in both mouse and human cells. This newly discovered, PTC-associated function of triamterene is independent of the epithelial sodium channel (ENaC) activity, the target of triamterene’s antihypertensive action. Importantly, after two weeks of administering 120 mg/kg triamterene to *IDUA*-W402X mice, GAGs were completely restored to normal levels in multiple tissues, including the brain and bone. Together, these data suggest that triamterene represents a promising new drug for treating MPS I-H patients who harbor a nonsense mutation.

## 2. Results

### 2.1. Screening Small Molecules to Identify New PTC Suppression Agents

To identify new compounds with the ability to suppress PTCs, we previously generated a NanoLuc reporter that harbors a point mutation at codon W134, resulting in the formation of a TGA premature termination codon (W134X) in the NanoLuc open reading frame [11] (Figure 1a). The NanoLuc cDNA was fused to a downstream human β-globin construct that contains intronic regions positioned >50 nucleotides downstream of the W134X PTC that elicit exon junction complex (EJC)-dependent NMD. This reporter is unique because it can identify not only molecules that stimulate readthrough, but also those that inhibit NMD (or alter mRNA abundance by other mechanisms), or dual-acting molecules that simultaneously promote readthrough and alter mRNA stability. The NanoLuc dual readthrough/NMD (RT/NMD) reporter was stably expressed in Fischer rat thyroid (FRT) cells, a cell line previously found to be responsive to readthrough [11]. This reporter cell line was used to screen a library of ~10,000 low molecular weight compounds, including many FDA-approved drugs, for those that increase NanoLuc activity.

Aminoglycosides have been shown to bind to a region of the ribosome known as the decoding center [14,15,16,17]. Previous studies reported that the presence of G418 may enhance the readthrough response to weak PTC suppression agents [11,13] or identify readthrough enhancer molecules that have no readthrough activity themselves but can greatly enhance the activity of aminoglycosides [12]. To help identify a wider selection of readthrough agents, we performed our screen with NanoLuc RT/NMD reporter FRT cells treated with each compound alone and in combination with a low dose (100 μg/mL) of G418. When administered alone, 11 compounds elicited a NanoLuc response greater than the G418 positive control response (see blue points lying above the dashed line) (Figure 1b). When co-administered with G418, a subset of compounds was identified that generated a >200-fold increase in NanoLuc activity when combined with G418 and resulted in a >20-fold increase above that obtained with G418 alone (Figure 1c). These results indicate strong, synergistic increases in reporter response. These results support our hypothesis that treating cells with G418 may render the translational machinery more sensitive to perturbation, allowing the identification of previously unidentified readthrough agents or molecules that enhance readthrough.

### 2.2. Triamterene Alleviates MPS I-H Biochemical Endpoints in IDUA-W402X MEFs

We next examined the top 17 molecules identified from the screen that produced the most robust readthrough when combined with G418 (Figure 1c; Appendix A) for the ability to restore α-L-iduronidase activity in immortalized *IDUA*-W402X mouse embryonic fibroblasts (MEFs). These MEFs were generated from homozygous *IDUA*-W402X MPS I-H mice that harbor a PTC homologous to the *IDUA*-W402X nonsense mutation, the most common mutation found in MPS I-H patients [18,19]. We previously found that the readthrough response in *IDUA*-W402X MEFs correlates well with the readthrough response in many MPS I-H mouse tissues [8,19,20]. Among the hits examined in MEFs, we found four that increased α-L-iduronidase activity well above basal levels when administered alone (Figure 2a). The increases provided by imatinib and vorinostat were marginal and these compounds were not addressed further. The two best-responding compounds included the FDA-approved drugs triamterene and doxazocin. Importantly, these compounds restored more α-L-iduronidase activity than G418 (Figure 2a). Further examination of doxazocin and triamterene showed that they both increased α-L-iduronidase activity in a dose-dependent manner (Figure 2b). However, in contrast to the NanoLuc reporter assays in FRT cells, no synergy was observed in *IDUA*-W402X MEFs between doxazocin (Appendix A) or triamterene (Appendix A) with either G418 or the non-aminoglycoside readthrough agent, PTC124 (also known as ataluren^®^ or translarna^TM^). This suggests that the synergistic increases in readthrough observed with the NanoLuc reporter may be dependent on factors such as the cell type or the PTC mRNA context.

We next surveyed whether triamterene or doxazocin restored enough α-L-iduronidase activity to generate a corresponding reduction of glycosaminoglycan (GAG) accumulation in *IDUA*-W402X MEFs [19,20] (Figure 2c). Using a GAG dye-binding assay, we found a dose-dependent decrease in sulfated GAG accumulation in *IDUA*-W402X MEFs treated with doxazocin or triamterene, with triamterene at a 30 μM dose reducing GAGs to wild-type (WT) levels (*p* = 0.8501). This indicates that the readthrough generated by triamterene or doxazocin provided a sufficient level of α-L-iduronidase activity to significantly reduce GAG accumulation. Similar amounts of α-L-iduronidase activity were previously shown to significantly reduce GAG accumulation in *IDUA*-W402X MEFs and mice treated with aminoglycosides [19,20]. This level of enzyme activity has also been reported to be associated with attenuated forms of MPS I in patients [10].

### 2.3. Triamterene Induces Readthrough and Stabilizes the IDUA-W402X mRNA

To examine whether the mechanism by which triamterene and doxazocin restored α-L-iduronidase activity is via suppression of translation termination at the *IDUA*-W402X nonsense mutation, we utilized previously characterized dual luciferase readthrough reporters [20]. These reporters are not subject to NMD, allowing us to determine the contribution from readthrough alone. The dual luciferase reporters consist of a cDNA expressing an upstream *Renilla* luciferase and a downstream firefly luciferase that are separated by an in-frame readthrough cassette containing a PTC or a WT codon (Figure 3a). We introduced the W402X mutation (TAG) or the WT tryptophan codon (TGG) flanked by three codons of upstream and downstream *IDUA* context into the readthrough cassette and stably expressed these reporters in HEK293 cells. Readthrough was measured as the firefly activity normalized to the *Renilla* activity in the PTC reporter cells relative to WT reporter cells. Using the dual luciferase reporter, we observed that triamterene generated a robust, dose-dependent increase in *IDUA*-W402X readthrough relative to basal levels. In contrast, no significant increase in readthrough was detected with doxazocin. This result indicates that triamterene, but not doxazocin, increases protein activity via readthrough of the UAG stop codon in the *IDUA*-W402X mRNA context.

We next examined whether triamterene or doxazocin altered steady-state *IDUA* mRNA abundance. We quantified *IDUA* mRNA in *IDUA*-W402X or WT MEFs using qPCR [8] (Figure 3b). In WT MEFs, triamterene did not significantly alter *IDUA* steady-state mRNA abundance compared to the vehicle alone control, while doxazocin led to a modest increase at a 60 μM concentration. However, in *IDUA*-W402X MEFs, which have much lower steady-state *IDUA* mRNA levels due to activation of NMD by the W402X mutation, triamterene increased *IDUA* mRNA abundance by 2.5- to 5-fold, while doxazocin increased *IDUA* mRNA levels by 2- to 3-fold. These results suggest that triamterene suppresses the *IDUA*-W402X mutation through a combination of readthrough and *IDUA* mRNA stabilization in a PTC-dependent manner, while doxazocin appears to work mainly by increasing *IDUA* mRNA levels in a PTC-independent manner.

To further examine the specificity of triamterene for PTCs, we next determined whether triamterene affects α-L-iduronidase activity and GAG levels in immortalized MEFs derived from an MPS I-H knockout mouse (KO). The KO mouse carries an insertional mutation in the mouse *IDUA* locus that is expected to be unresponsive to readthrough [22,23]. Slight increases of α-L-iduronidase were observed in triamterene-treated KO MEFs, but less than the increases observed in triamterene-treated *IDUA*-W402X MEFs (Figure 4a). Importantly, GAG analysis results correlated well with the α-L-iduronidase assay, where a large decrease in GAGs was observed in triamterene-treated *IDUA*-W402X MEFs, while little to no reduction in GAGs was observed in treated KO MEFs (Figure 4b). These results suggest that triamterene restores α-L-iduronidase primarily through a PTC-dependent mechanism.

By blocking the epithelial sodium channel (ENaC) in the distal kidney, triamterene acts as a potassium-sparing diuretic that is used to treat hypertension and edema, another well-known ENaC inhibitor is amiloride [24], which has a chemical structure that is distinct from triamterene. We tested whether amiloride, via its ENaC-blocking function, also affected α-L-iduronidase activity and GAG levels in *IDUA*-W402X MEFs. Unlike triamterene, amiloride did not generate a significant dose-dependent increase in α-L-iduronidase activity (Figure 4c). Furthermore, amiloride did not reduce GAGs in *IDUA*-W402X MEFs (Figure 4d). These data suggest that triamterene moderates MPS I-H biochemical endpoints through a mechanism that is independent of its role as an ENaC inhibitor. Analysis of thiazide, another diuretic that is clinically co-administered with triamterene and marketed as dyazide^TM^, did not significantly affect α-L-iduronidase activity in *IDUA*-W402X MEFs (Appendix A). Furthermore, thiazide did not inhibit the ability of triamterene to rescue α-L-iduronidase activity.

To ensure that the positive effect of triamterene on mouse MPS I-H biochemical endpoints were also possible in human cells, we examined the effect of triamterene on α-L-iduronidase activity in induced pluripotent stem cells (iPSCs) that were reprogrammed from a skin fibroblast cell line obtained from an MPS I-H patient heterozygous for the *IDUA* p.Q70X and p.W402X nonsense mutations [25]. After treating these iPSCs for 72 h with triamterene, we found a significant 1.5-fold increase in α-L-iduronidase specific activity compared to iPSCs treated with vehicle alone (Figure 4e). No change in α-L-iduronidase activity was observed in similarly treated WT iPSCs.

We next examined whether the increase in α-L-iduronidase in triamterene-treated p.Q70X/p.W402X iPSCs could reduce GAG levels. Consistent with other studies examining iPSCs derived from MPS I-H patients [26], we found that total sulfated GAGs were elevated ~2-fold in MPS I-H iPSCs compared to the WT control (Figure 4f). We found no significant change in GAG levels in WT iPSCs treated with triamterene compared to the vehicle control. However, we found that triamterene treatment of MPS I-H iPSCs significantly reduced GAGs compared to the vehicle-treated control. Furthermore, the highest 15 μM dose of triamterene reduced GAGs in the MPS I-H iPSCs to WT levels. This suggests that triamterene alters biochemical endpoints similarly via a PTC-dependent manner in both mouse and human MPS I-H models.

### 2.4. In Vivo Administration of Triamterene Normalizes GAG Levels in IDUA-W402X Mice

We next examined whether triamterene alleviated MPS I-H biochemical endpoints in vivo using homozygous *IDUA*-W402X mice, which recapitulate most of the same abnormalities observed in MPS I-H patients and respond to readthrough [8,18,19]. We administered triamterene to eight-week-old *IDUA*-W402X mice via once-daily oral gavage for a total of two weeks (14 doses). After treatment was completed, sulfated GAG levels were measured in mouse tissues using a GAG dye-binding assay [19,20]. For all tissues examined, we found triamterene treatment led to a dose-dependent decrease in GAGs in treated *IDUA*-W402X mice relative to vehicle-treated controls (Figure 5). At the 120 mg/kg dose, triamterene reduced GAGs to WT levels in all tissues examined. However, administration of 120 mg/kg triamterene to *IDUA*-KO mice for two weeks did not lead to a reduction in GAG storage (Appendix A), further confirming that triamterene also alleviated in vivo MPS I-H biochemical endpoints via a PTC-dependent mechanism. Longer-term studies to examine whether triamterene attenuated MPS I-H progression in *IDUA*-W402X mice were unsuccessful. This negative result is likely due to its greater metabolism in mice since triamterene was found to be much less stable in mouse liver microsomes than in human liver microsomes (Appendix A). This suggests that this higher level of metabolism of triamterene likely inactivated its ability to mediate PTC suppression in mouse tissues following extended treatment periods (Appendix A).

Urine GAG levels have previously been used as a marker for both diagnosis of MPS I-H [27] as well as an endpoint to assess the effectiveness of treatments for MPS I-H, such as enzyme replacement therapy [28] and hematopoietic stem cell transplantation [29]. We collected urine samples from 8-week-old *IDUA*-W402X mice before treatment and at 2-week intervals during an 8-week treatment regimen in which triamterene was administered once daily at 120 mg/kg. We determined the levels of dermatan sulfate (DS) and heparan sulfate (HS) specifically in these urine samples by mass spectrometry. We found that DS (Figure 6a) and HS (Figure 6b) in *IDUA*-W402X mice were elevated ~10-fold and 44-fold, respectively, compared to age-matched WT mice before treatment (8-week-old mice). In contrast, the level of chondroitin sulfate (Figure 6c), which is not an α-L-iduronidase substrate, was similar in both WT and *IDUA*-W402X controls. Over the course of treatment, the DS decreased between 19–35% in triamterene-treated *IDUA*-W402X mice, with a statistical difference in DS levels between the treated and control mice reached at 2-weeks of treatment. HS was similarly decreased between 18–32% in triamterene-treated *IDUA*-W402X mice compared to the vehicle-treated controls, with differences between the groups achieving statistical significance at both 2- and 4-weeks of treatment (Figure 6a–b), but not at later time points. Importantly, chondroitin sulfate levels did not significantly change among any of the cohorts during the 8-week treatment period (Figure 6c). These results indicate that triamterene specifically reduced the GAG substrates for α-L-iduronidase, corroborating our previous data that showed triamterene rescues α-L-iduronidase function and reduces GAG accumulation via readthrough of PTCs. The diminished GAG reduction after 2- to 4-weeks of treatment is consistent with mouse liver microsome studies showing extensive metabolism of triamterene (as shown in Appendix A), likely reducing the efficacy of triamterene to promote PTC suppression over longer-term treatments in this mouse model.

## 3. Discussion

Standard of care treatments for MPS I-H includes hematopoietic stem cell transplantation (HSCT), and enzyme replacement therapy (ERT). Both treatments help alleviate some MPS I-H symptoms and extend the patient’s lifespan. However, Aldurazyme^TM^, a recombinant form of α-L-iduronidase that is administered as an ERT, cannot cross the blood/brain barrier to prevent the onset of neurological deterioration. While HSCT can halt the progression of neurological phenotypes if administered early (<16-months of age) [30], it carries significant inherent safety risks. The rate of MPS I-H patients surviving an initial HSCT with efficient engraftment has been reported to be around 70% [31]. In addition, neither ERT nor HSCT prevent the onset of MPS I-H manifestations in the heart valves, cornea, or bone, which significantly contribute to the morbidity and mortality of MPS I-H patients [32]. Safer, more effective treatments for MPS I-H are needed.

Nonsense suppression therapy represents a promising approach to treating MPS I-H patients who carry a PTC. Notably, more than 70% of all MPS I-H patients carry a nonsense mutation [9]. In addition, the threshold for phenotypic improvement of MPS I-H is quite low, with as little as 0.3% of normal α-L-iduronidase activity differentiating between patients with the severe, Hurler form of MPS I from patients from the attenuated, Scheie form [10], who have a normal intellect and life expectancy. Furthermore, readthrough agents are low molecular weight compounds that may access tissues that are impenetrable to current MPS I-H treatments, such as the brain and bone [19,20].

In the current study, we implemented several strategies to identify new, effective readthrough agents. First, we used a NanoLuc-based dual RT/NMD reporter that allowed us to identify not only molecules that stimulate PTC readthrough, but also compounds that stabilize the PTC-containing mRNA by inhibiting NMD or via other mechanisms such as passive mRNA stabilization [33]. An advantage of this reporter is that it can be used to find compounds (or combinations of compounds) that target multiple PTC suppression mechanisms, a strategy likely to generate more protein function than targeting only a single readthrough mechanism [6,8]. Furthermore, we screened each compound alone and combined with a low dose of the aminoglycoside, G418. The presence of G418 during screening enhanced readthrough, allowing us to identify greater numbers of compounds with potential readthrough activity, or with the ability to enhance the activity of other readthrough compounds as previously reported [12]. Triamterene was identified as a hit from our screen that alone stimulated readthrough, and in combination with G418, produced a synergistic increase in readthrough in the primary reporter assay screen. However, we did not observe synergy with triamterene and G418 in *IDUA*-W402X MEFs. We speculate that certain aspects of translation and/or mRNA turnover might differ between embryonic and differentiated cells which could affect whether synergy occurs. For example, we have found that the extent by which *IDUA*-W402X mRNA abundance is reduced by NMD differs between mouse embryonic tissue and tissues from adult mice. *IDUA*-W402X mRNA abundance is reduced to around 5% of wild-type levels in MEFs, while in tissues from adult mice, *IDUA*-W402X mRNA abundance is reduced to 25–50% of wild-type [8]. This may be at least one factor that contributes to the lack of synergy in MEFs.

In both in vitro and in vivo *IDUA*-W402X MPS I-H models, triamterene restored α-L-iduronidase activity and decreased GAG storage in a dose-dependent manner. Unlike ataluren [34] or the synthetic aminoglycoside NB84 [19,20], which only partially attenuated GAG accumulation in most *IDUA*-W402X mouse tissues, a 120 mg/kg dose of triamterene for 2-weeks completely normalized GAG levels in all the tissues examined and significantly reduced DS and HS in urine samples from *IDUA*-W402X mice. Even the partial GAG reductions previously observed in NB84-treated *IDUA*-W402X mice significantly attenuated the progression of MPS I-H [19]. Given that triamterene normalized GAG levels in all *IDUA*-W402X mouse tissues, it was anticipated that triamterene would also prevent or greatly attenuate the progression of MPS I-H in *IDUA*-W402X mice. Unfortunately, this result was not observed; microsome stability assays suggested that triamterene likely undergoes extensive metabolism in mouse liver (Appendix A), which appears to inactivate its PTC suppression activity. This finding precluded examination of the effects of long-term triamterene treatment in *IDUA*-W402X mice. However, this should not be the case in humans since triamterene is much more stable in human microsomal extracts and is routinely used as a diuretic in patients for extended periods of time.

Additional in vitro studies in our *IDUA*-W402X MEFs indicated that triamterene restores α-L-iduronidase primarily through PTC-dependent mechanisms that at least entail promoting readthrough of the *IDUA*-W402X PTC and stabilizing the *IDUA*-W402X mRNA. Furthermore, the PTC suppression activity mediated by triamterene is independent of ENaC function. The ability of triamterene to suppress PTCs in human cells was demonstrated in HEK293 readthrough reporter cells and in MPS I-H patient iPSCs, where triamterene restored sufficient α-L-iduronidase activity to normalize GAG accumulation.

Together, this evidence suggests that we have identified a novel function for triamterene as a PTC suppression agent. Because triamterene is already an FDA-approved drug, repurposing triamterene as a nonsense suppression compound could quickly allow triamterene to enter clinical trials as a readthrough therapy for MPS I-H patients. While the 120 mg/kg dose administered to *IDUA*-W402X mice is significantly higher than the clinical dose normally administered to patients, triamterene is metabolized much more extensively in mouse liver microsomes than in human liver microsomes (Appendix A), suggesting that the triamterene dose that normalized GAGs in *IDUA*-W402X mice may be comparable to the range of triamterene dosing that is used clinically. Since ENaC function does not influence the readthrough of the *IDUA*-W402X mutation, it may also be possible that modifying the structure of triamterene to increase its potency as a readthrough agent while eliminating its effect on ENaC can reduce potential side effects associated with increased potassium levels. Similar medicinal chemistry approaches have generated safer, more effective aminoglycosides for readthrough [20,35,36,37]. Altogether, these data suggest that triamterene (or triamterene-like) molecules may be effective readthrough agents for treating MPS I-H. Triamterene could potentially be used as a readthrough agent for other disease models as well. While we mainly focused on the *IDUA*-W402X context, it is important to note that triamterene alone increased readthrough by 13-fold in the NanoLuc reporter context. The NanoLuc context (UGA) is significantly different than the *IDUA*-W402X context (UAG), suggesting that triamterene likely induces readthrough in a variety of different contexts.

## 4. Materials and Methods

### 4.1. Cell Culture

Mouse embryonic fibroblasts (MEFs) were immortalized using the SV40 large T antigen as previously described [8,20]. MEFs utilized in this study were derived from homozygous wild-type mice (WT), knock-in MPS I-H mice homozygous for a genomic point mutation that generates a PTC homologous to the *IDUA* p.W402X nonsense mutation (*IDUA*-W402X) [18], and knock-out MPS I-H mice carrying an insertional mutation in the mouse *IDUA* locus (*IDUA*-KO) [22]. MEF cell lines were cultured at 37 °C with 6.5% CO_2_ in Dulbecco’s Modification of Eagle’s Medium containing 4.5 g/L glucose, L-glutamine and sodium pyruvate (Corning Cellgro 10-013-CV, Manassas, VA, USA). This media was supplemented with 100 units/mL penicillin/streptomycin (Corning Cellgro 30-002-Cl), MEM non-essential amino acids (Corning Cellgro 25-025-Cl) at a final concentration of 1% (*v*/*v*), and fetal bovine serum (Atlanta Biologicals S11150, Flowery Branch, GA, USA) at a final concentration of 10% (*v*/*v*). HEK293 cells were cultured similarly. Fisher rat thyroid (FRT) cells were cultured in Nutrient Mixture F-12 Coon’s modification media (Sigma F6636, St. Louis, MO, USA) supplemented with 5% fetal bovine serum. For stably transfected HEK293 and FRT cells, 100 units/mL penicillin/streptomycin was added to the media in the absence of zeocin to prevent bacterial contamination. iPSCs were reprogrammed from MPS I-H patient skin fibroblasts [25] as previously described [38,39]. iPSCs were cultured in mTeSR Plus media (Stemcell Technologies #100-0276, Vancouver, BC, Canada) on plates coated with Geltrex (Gibco #A14133-02, Waltham, MA, USA) according to manufacturer instructions.

### 4.2. Construction of the Readthrough Reporters

The NanoLuc readthrough/NMD (RT/NMD) reporter was constructed as previously described [11]. The NanoLuc open reading frame was obtained from the pFN[Nluc/CMV/neo] plasmid (Promega CS181701, Madison, WI, USA). Initially, a multi-cloning site was placed into the Xho I/Not I sites of this plasmid using the annealed oligos DB4078 [5′-tcgagccaag cttgcatgcct gcaggtcgact ctagaggatcc ccggggaattcgc-3′] and DB4079 [5′-ggccgcgaat tccccgggga tcctctagag tcgacctgcag gcatgcaagcttggc-3′] to remove the barnase sequence and simplify subsequent cloning. This new construct (pDB1333) was used as a template to generate the W134X (UGA) premature termination codon in the NanoLuc gene (pDB1345) using site-directed mutagenesis with the forward primer DB4144, 5′-gggaccctgt gaaacggcaac-3′ and the reverse primer DB4145, 5′-gttgccgttt cacagggtccc-3′. The final W134X NanoLuc RT/NMD reporter was generated by replacing the *Renilla* gene within the *Renilla*-β-globin/pcDNA3.1Zeo(−) plasmid (pDB1329) with the W134X NanoLuc gene from pDB1345 using Nhe I/Xho I sites such that the NanoLuc gene was fused in-frame with exon 1 of β-globin.

The construction of the *IDUA*-W402X dual-luciferase reporters has been previously described [20]. The original p2luc dual luciferase readthrough constructs were a gift from Dr. John Atkins [40]. The p2luc construct was modified to express either the wild-type (WT) mouse *IDUA* codon (UGG) that is homologous to the W402 codon in the human *IDUA* cDNA, or the *IDUA* W402X premature stop codon (UAG), along with three codons of upstream and downstream mouse *IDUA* sequence. Complementary oligonucleotides for generating the *IDUA*-W402X construct: [5′-tcgacg gaacaa ctctag gcagag gtcg-3′; 5′-gatccg acctct gcctag agttgt tccg-3′], and the WT *IDUA* construct: [5′-tcgacg gaacaa ctctgg gcagag gtcg-3′; 5′-gatccg acctct gcccag agttgt tccg-3′] were annealed to generate double-stranded DNA fragments that were ligated into the Sal I and BamH I restriction sites of the p2luc vector, yielding the *IDUA*-W402X (pDB1134), and *IDUA*-WT (pDB1133) p2luc constructs. Using the Not I and Nhe I restriction sites, the *IDUA*-W402X and *IDUA*-WT dual luciferase constructs were subcloned into pcDNA3.1Zeo(+) to generate pDB1325 and pDB1326, respectively, for stable expression in mammalian cells.

### 4.3. Generation of Readthrough Reporter Cells

The W134X NanoLuc RT/NMD reporter was linearized with BgI II and transfected into Fisher rat thyroid (FRT) cells using Lipofectamine 2000 (Invitrogen 11668, Austin, TX, USA). Stably transfected FRT cells were selected by the addition of 800 μg/mL zeocin to the growth media for 2–3 weeks. Stable FRT reporter cell lines were maintained by the addition of 200 μg/mL of zeocin to the media. Monoclonal FRT cell lines were established by collecting cells from single-cell-derived colonies and expanding them.

The *IDUA*-W402X and *IDUA*-WT dual luciferase constructs (pDB1325 and pDB1326, respectively) were linearized with Bgl II and transfected into human embryonic kidney (HEK) 293 cells, which were cultured in the presence of 200 μg/mL of zeocin for 2–3 weeks to select for cells stably expressing the reporters. Reporter HEK293 cells were also maintained in 200 μg/mL of zeocin.

### 4.4. High Throughput Screening Assay

This screen was similarly conducted with the W134X NanoLuc RT/NMD-FRT Monoclonal #14 cells as previously described [11]. The reporter cells were cultured in Coon’s F-12 Medium (Cedar Lanes Labs Catalog #F 0855, Burlington, ON, Canada) supplemented with 5% heat-inactivated fetal bovine serum (Omega Scientific Catalog #FB-11, Tarzana, CA, USA) and 1% L-glutamine, harvested, and then re-suspended at 400,000 cells/mL in media supplemented with 5% heat-inactivated fetal bovine serum, 1% Pen/Strep/Glut (Corning Catalog #30-009CI) and 1% HEPES (Gibco Catalog #15630-080). Using Beckman Coulter FX, 5 µL of compounds/controls were added to wells of 384-well white, opaque bottom plates (Corning Catalog #3570BC). Control and screening compounds were diluted in media with DMSO (Sigma-Aldrich Catalog #D8418) at 6× concentration. The wells of columns 1 & 2 were used as the negative (cell) control and contained 5 µL of media with DMSO. The wells of columns 23 & 24 were used as the positive control and contained 5 µL of G418 (Corning Catalog #30-234-CI) diluted in media with DMSO. 5 µL of screening compounds were added to the wells of columns 3–22. Using Thermo Matrix wellmate dispenser, 25 µL of W134X cells were added to the wells of columns 1, 2, 23, & 24 of the assay plates. For test wells, 25 µL of W134X cells were added to the wells of columns 3–22. For the screening condition in the presence of G418, this reagent was added to the cells before dispensing to give a final concentration of 100 µg/mL. The final concentration of cells was 10,000 cells per well with a final assay volume of 30 µL and a final concentration of DMSO concentration of 0.6%. G418 was used as the positive control condition with a final concentration of 300 µg/mL. Compounds from 2 libraries were tested on this screen. Library compounds purchased from Enamine (Monmouth Junction, NJ, USA) were screened at 30 µg/mL final concentration. A library of FDA-approved compounds was screened at 60 µM final concentration. The assay plates were gently tapped and placed in a 37 °C with 5% CO_2_, high humidity incubator for 48 hrs. After incubation, the plates were cooled to room temperature. Using the Biomek Benchtrak robot, 30 µL of room temperature NanoGlo reagent (Promega Catalog #N1150) diluted 1:50 was added to all the wells of the assay plate. After 10 min, the luminescence was measured using a Perkin Elmer Envision reader. Cell toxicity was monitored using a Celltiter-Glo Luminescent Cell Viability Assay (Promega #G7570).

### 4.5. Luciferase Readthrough Assays

Equal numbers of the W134X NanoLuc RT/NMD monoclonal reporter cells were seeded into 96-well plates and cultured without zeocin for 24 h. The cells were then treated with screening compounds for 24 h. Dimethyl sulfoxide (DMSO) served as the vehicle in which compounds were administered at a final concentration of 60 μM or 30 μg/mL. After treatment, the cells were washed once with phosphate-buffered saline (PBS), followed by incubation with passive lysis buffer (PLB) (Promega, E1941) for 20 min at room temperature. NanoLuc activity in cell lysates was assayed with the Nano-Glo Luciferase Assay System (Promega, N1120) using the GloMax^®^ Discover System (Promega). NanoLuc activity was normalized to the lysate protein concentration, which was determined by the Bio-Rad Protein Assay Reagent (Bio-Rad 500-006, Hercules, CA, USA).

HEK293 dual luciferase reporter cells were similarly cultured, treated with screening compounds, and lysed as described above. Dual luciferase activity was assayed with the Dual Luciferase Assay System (Promega) using the GloMax Multi Detection System (Promega). The percent readthrough was calculated as the ratio of firefly/*Renilla* luciferase units expressed from the W402X construct relative to the WT construct × 100.

### 4.6. α-L-iduronidase Activity Assay

MEFs or iPSCs were seeded into 6-well culture dishes at a density of 1.5 × 10^5^ cells per well. Cells were grown to 50% confluency and then treated with screening agents for 48 h. Cells were subsequently washed with PBS and lysed in Mammalian Protein Extraction Reagent (Thermo Scientific P178501, Waltham, MA, USA) containing a protease inhibitor cocktail (Complete Roche 11873580001, Indianapolis, IN, USA). The total protein concentration was determined using the Bio-Rad Protein Assay. Approximately 75 μg of total lysate protein were incubated in a 70 μL reaction containing 0.12 mM 4-methyl-umbelliferyl-α-L-iduronide (Gold Biotech M-570-5, Saint Louis, MO, USA) and 0.42 mg/mL of D-saccharic acid 1,4-lactone monohydrate (a β-glucuronidase inhibitor) (Sigma S0375) in 130 mM sodium formate buffer, pH 3.5. The reaction was incubated for 72 h at 37 °C and then quenched with 300 μL of glycine buffer, pH 10.8. 200 μL of each sample was transferred to a 96-well plate (Corning 3631) and fluorescence was measured at an excitation = 365 nm and an emission = 450 nm using the GloMax^®^ Discover System (Promega). Free acid 4-methylumbelliferone (FMU) (Sigma M1381) in glycine buffer was used to generate a standard curve. Specific activity was calculated as nanomoles of FMU released per microgram of protein per hour. α-L-iduronidase activity remained linear over the 72-h incubation time with the substrate concentration remaining in excess over the incubation period. As shown in Appendix A, a standard curve was also established with the fluorescent 4-methyl-umbellyferone substrate in the presence of triamterene. Fluorescence associated with triamterene occurred only at high concentrations, which was not expected to be present in cell or tissue lysates with the concentrations used to treat MEFs or mice.

### 4.7. Glycosaminoglycan Quantitation in Cultured Cells

MEFs or iPSCs were seeded into 6-well culture dishes at a density of 5 × 10^4^ cells per well. Cells were grown to 50% confluency and then treated with screening agents for 48 h. Subsequently, the media was removed and the cells were scraped into 500 µL of 0.2 mg/mL papain in pH 6.5 phosphate buffer containing 0.6 mg/mL cysteine. The cell lysates were incubated in the papain solution at 65 °C with gentle agitation for 3 h. Samples were briefly microfuged for 10 min at 10,000× *g* at room temperature to remove debris. GAG levels were determined using the Blyscan Sulfated GAG Assay (Biocolor Ltd., Carrickfergus, UK). Briefly, 50 µL of each supernatant was mixed with 500 µL of the Blyscan Dye Reagent to bind sulfated GAGs. The dye-bound GAGs were pelleted by microfuging for 10 min at 10,000× *g* at room temperature. 500 µL of the Blyscan Dye Dissociation Reagent was added to each sample to dissociate the GAGs from the dye. 200 µL of each sample was transferred to a 96-well plate (Corning 3631) and the absorbance was measured at a wavelength of 650 nm absorbance using a GloMax^®^ Discover System (Promega). The total amount of sulfated GAGs precipitated from each sample was determined from a chondroitin 4-sulfate (Sigma C9819) standard curve. The total protein concentration in each lysate was determined using the Bio-Rad Protein Assay (5000006) from a standard curve generated using bovine serum albumin. The GAG levels are expressed as nanograms of GAGs per microgram of total protein. Standard curves for the GAG dye-binding assay using chondroitin 6-sulfate (Sigma C4384) (Appendix A), dermatan sulfate (EMD Biosciences 263301) (Appendix A), and heparan sulfate (Sigma-Aldrich H7640) (Appendix A) were performed in the absence and presence of triamterene (at concentrations 10- to 20-fold greater than the typical patient target serum concentrations with oral dosing [41]) to confirm that triamterene does not interfere with the binding of any of the different types of GAGs in the dye-binding assay.

### 4.8. Quantitation of Steady-State IDUA mRNA Levels

Total RNA was isolated from MEFs using the Ambion RiboPure kit (Fisher Scientific AM1924) and DNase-treated using the Turbo DNA-Free kit (Fisher Scientific AM1907). Polyadenylated RNA was reverse transcribed into cDNA in a 50 μL reaction containing 1 μg of total RNA; 0.5 mg/mL oligo dT; 1.2 mM dNTPs; 40U RNasin (Promega N2511); 10 mL of 5× AMV RT buffer and 40U AMV reverse transcriptase (Promega PM-M9004). RT reactions were incubated at 42 °C for 1.5 h, and then heat-inactivated at 65 °C for 15 min. The cDNA was ethanol precipitated and subjected to qPCR in a 25 μL reaction containing 12.5 μL iQ SYBR Green Supermix (Bio-Rad 1708882); 0.2 mM of each forward and reverse primer; and 2 μg of cDNA. The following primer sets (forward = Pf and reverse = Pr) were used: *IDUA* Pf: 5′-tgacaa tgcctt cctgag ctacca-3′ and *IDUA* Pr: 5′-tgactg tgagta ctggct ttcgca-3′; *Gapdh* Pf: 5′-ttccag tatgac tccact cacgg-3′ and *Gapdh* Pr: 5′-tgaaga caccag tagact ccacgac-3′; *Rpl13a* Pf: 5′-atgaca agaaaa agcggatg-3′ and *Rpl13a* Pr: 5′-cttttc tgcctg tttccgta-3′. The qPCR was performed using the CFX96 Real-Time PCR Detection System (Bio-Rad) using a program that included an initial 3-min denaturation step at 95 °C followed by 40 repeated cycles of a 10 s denaturation step at 95 °C and a 30 s annealing/extension step at 55 °C. Melt curve analysis was initially performed with each primer set to verify that only one gene product was generated from the PCR reactions. A standard curve was also performed using each primer set to ensure that under the PCR conditions used, the efficiency ranged between 90–110%. The average quantification cycle (Cq) was determined for each mRNA, and mRNA quantification was performed using the Livak (DDCq) method [21], where *Gapdh* and *Rpl13a* served as normalization controls. Cq values among the different samples for the various transcripts ranged from 10–34. The qPCR was performed using at least 8–12 replicates for each gene product from each sample.

### 4.9. Microsome Stability Assay

The amount of unaltered compound was determined in human and mouse liver microsomes (0.5 mg/mL protein) at 37 °C with 1.0 µM analyte. The reaction was quenched with acetonitrile after 0, 5, 10, 20, 30, and 60 min. The samples were then analyzed by LC-MS/MS methodology as previously described, with diclofenac and midazolam used as positive controls [42,43].

### 4.10. Animal Treatment

Triamterene was suspended in a vehicle composed of 0.5% methylcellulose in an artificially sweetened Kool-Aid solution. This mixture was administered orally to 8- to 10-week-old mice via once-daily gavage for a total of 14 days. Doses ranged from 30–120 mg/kg. Control mice were administered the Kool-Aid vehicle alone. All animals were provided standard chow and water ad libritum. All animal work was conducted according to relevant national and international guidelines. All animal protocols used in this study were reviewed and approved by the UAB IACUC (protocol IACUC-20364).

### 4.11. GAG Quantitation in Mouse Tissues

This assay was performed as previously described [8,19,20]. Tissues were homogenized using a Tissue Tearor homogenizer in chloroform: methanol (2:1 *v*/*v*). Defatted tissue was dried in a speedvac, weighed, and then suspended in 100 mM dibasic sodium phosphate, pH 6.5 containing 0.6 mg/mL cysteine and 2 mg/mL papain (Sigma P4762). The mixture was digested at 60 °C for 18–24 h with constant agitation. The samples were then microfuged at 10,000× *g* for 15 min and the supernatant was used to quantitate the tissue GAGs using the Blyscan Sulfated GAG Assay (Biocolor Ltd., Carrickfergus, UK). The total amount of sulfated GAGs precipitated from each sample was determined from a standard curve using chondroitin 4-sulfate (Sigma C9819). The GAG levels are expressed as micrograms of GAGs per milligram of defatted, dried tissue.

### 4.12. GAG Determination in Mouse Urine

Urine was collected from individual mice via spontaneous urination into a weigh boat, filtered using a 0.22 μm syringe filter, and then stored at −80°C until assay. Urinary glycosaminoglycans (chondroitin sulfate, dermatan sulfate, and heparan sulfate) were quantified using UPLC MSMS as previously described [44]. Briefly, GAGs were digested using methanolic HCl into three unique disaccharides that are specific for chondroitin sulfate, dermatan sulfate, or heparan sulfate. The dimers were separated by ultra-performance liquid chromatography (UPLC) and analyzed by electrospray ionization tandem mass spectrometry (MS/MS) using selected reaction monitoring. Calibration curves were generated using commercially available standards (Sigma).

### 4.13. Statistics

All statistics were calculated with ANOVA models using Prism 9 software. Sample sizes are indicated in figure legends and/or the materials and methods and were selected for each assay based on previous data. Among the biochemical endpoints, the GAG assay was the most variable and 8 mice per group were required to detect a 20% decrease in brain GAGs (with 85% power at a 0.05 significance level). We will therefore include 8 mice per cohort to ensure that our sample size supports adequate statistical power.

## Figures and Tables

**Figure 1 ijms-24-04521-f001:**
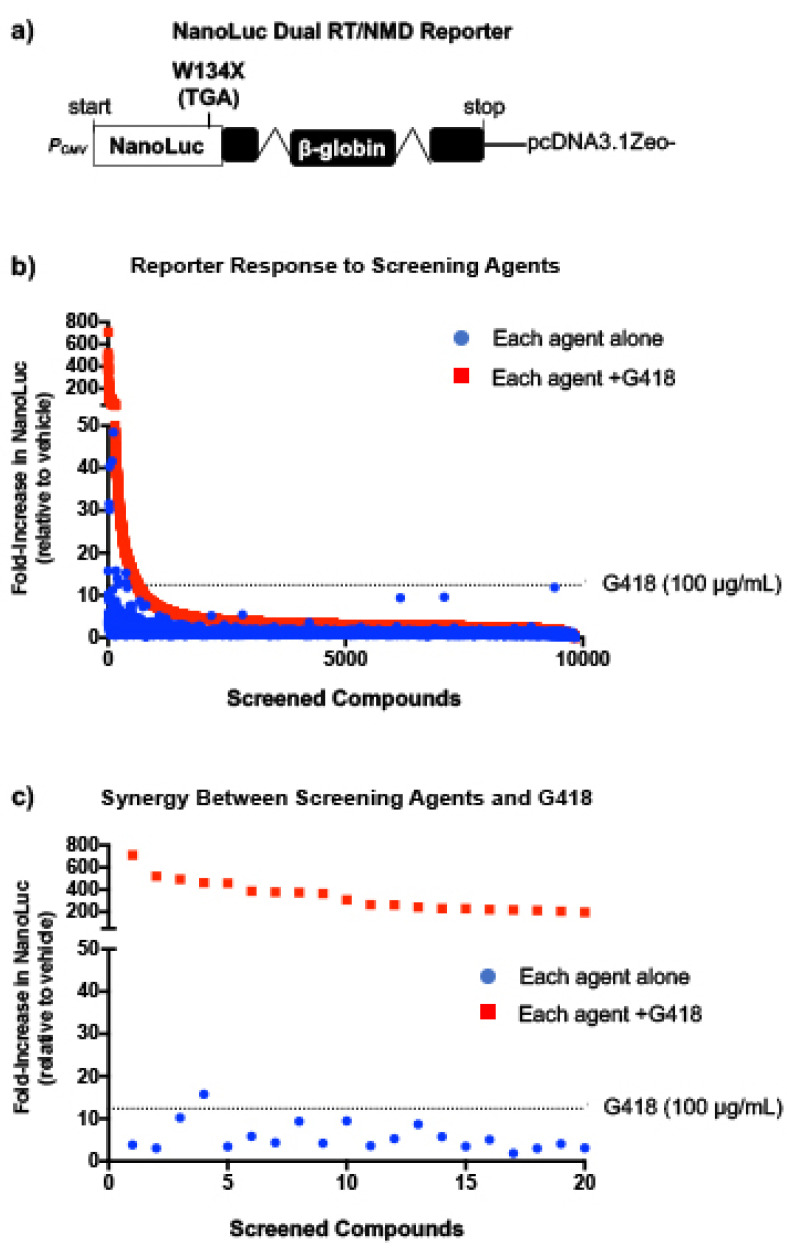
Identifying compounds from a high throughput screen (HTS) of 10,000 low molecular weight compounds for those that suppress PTCs. (**a**) The NanoLuc dual RT/NMD reporter shown was stably expressed in Fischer rat thyroid (FRT) cells, which were used to screen a library of ~10,000 low molecular weight compounds at a single concentration (60 μM or 30 μg/mL) for those that increase NanoLuc activity via stimulating readthrough and/or elevating PTC-containing mRNA abundance (via NMD inhibition or other mechanisms). (**b**) Fold-increase in NanoLuc activity in response to each of the 10,000 compounds alone (blue) or in the presence of 100 μg/mL G418 (red), relative to the DMSO vehicle control. (**c**) The 20 compounds produced the greatest fold increase in NanoLuc reporter response when combined with G418. In panels (**b**,**c**), the dotted line represents the NanoLuc reporter response to treatment with 100 μg/mL G418 alone.

**Figure 2 ijms-24-04521-f002:**
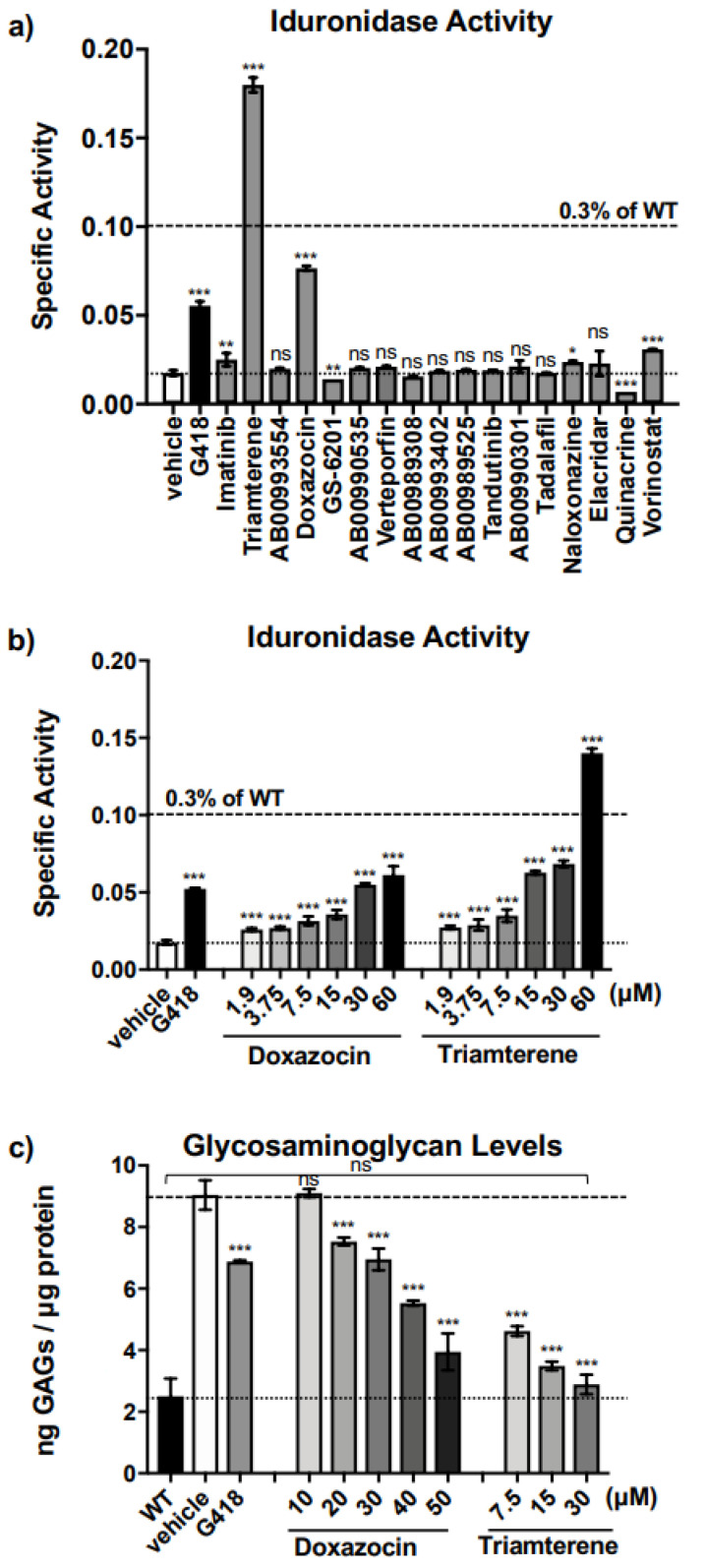
Examination of primary HTS hits for the ability to restore α-L-iduronidase activity in *IDUA*-W402X MEFs. (**a**) *IDUA*-W402X MEFs were cultured in the presence of the top 17 screening hits at a concentration of 60 μM or 30 μg/mL (Appendix A) for 48 h. α-L-iduronidase specific activity was measured in cell lysates using a fluorescent substrate as described in Section 4 [20]. Each column represents the mean ± SD of a representative experiment (n = 4–6). Hits that begin with the letters “AB” are proprietary compounds of Southern Research. Samples treated with vehicle or G418 alone serve as negative and positive controls, respectively. Dose responses of (**b**) α-L-iduronidase specific activity (nanomoles of substrate released per μg of protein per hour) [20], and (**c**) sulfated GAG levels [20] in *IDUA*-W402X MEFs treated with doxazocin or triamterene. Each column represents the mean ± SD of two independent experiments (n = 6–8). In panels (**a**,**b**), the upper dashed line represents 0.3% of WT α-L-iduronidase activity, a level reported to alleviate MPS I-H phenotypes in *IDUA*-W402X mice [19]; the lower dashed line is the activity found in the negative control. In panel c, the upper dashed line represents the GAG level in homozygous *IDUA*-W402X MEFs, while the lower dashed line represents the GAG level in WT MEFs. For panels (**a**–**c**), samples treated with vehicle and G418 serve as negative and positive controls, respectively. Exact *p* values were calculated using a one-way ANOVA with Dunnett’s multiple comparisons tests where values for treated cells were compared to the vehicle control or the WT control as indicated. *** *p* < 0.0001; ** *p* < 0.001; * *p* < 0.05; ns indicates *p* > 0.05.

**Figure 3 ijms-24-04521-f003:**
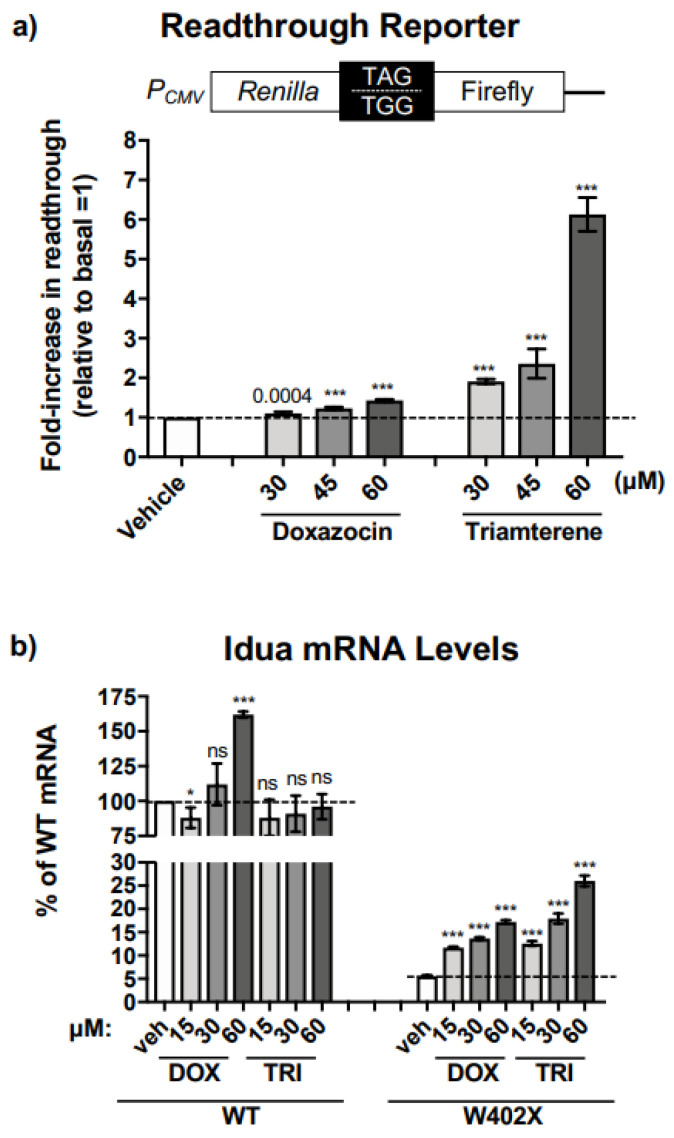
Triamterene mediates readthrough of the *IDUA*-W402X mutation and increases steady-state *IDUA*-W402X mRNA. (**a**) HEK293 cells stably expressing a dual luciferase reporter containing the *IDUA*-WT or *IDUA*-W402X contexts were cultured in the presence of doxazocin or triamterene for 24 h prior to assay. The level of W402X readthrough was calculated as the firefly/*Renilla* units for the TAG (W402X) construct relative to the firefly/*Renilla* units for the TGG (WT) control. The data is expressed as the fold-increase in the % readthrough (FF/Renilla × 100) for treated cells relative to the vehicle-treated control. Each column represents the mean ± SD of two independent experiments (n = 8–12). (**b**) *IDUA*-W402X MEFs were cultured in the presence of doxazocin or triamterene for 48 h. Total RNA was isolated and polyadenylated mRNA was reverse transcribed to cDNA and subjected to quantitative PCR to determine the steady-state *IDUA* mRNA abundance [8]. The Livak quantitation method [21] was used to quantify *IDUA* mRNA levels, with *Gapdh* serving as the normalization control. The *IDUA* mRNA levels in all experimental samples are expressed as the percent of the *IDUA* mRNA level in the WT vehicle control = 100. The dashed lines indicate the vehicle alone controls. Each column represents the mean ± SD of *IDUA* levels from two independent experiments (n = 8–12). *p* values were calculated using a one-way ANOVA with Tukey’s multiple comparisons tests where values for treated cells were compared to the vehicle control or the WT control as indicated. *** *p* < 0.0001; * *p* < 0.05; ns indicates *p* > 0.05.

**Figure 4 ijms-24-04521-f004:**
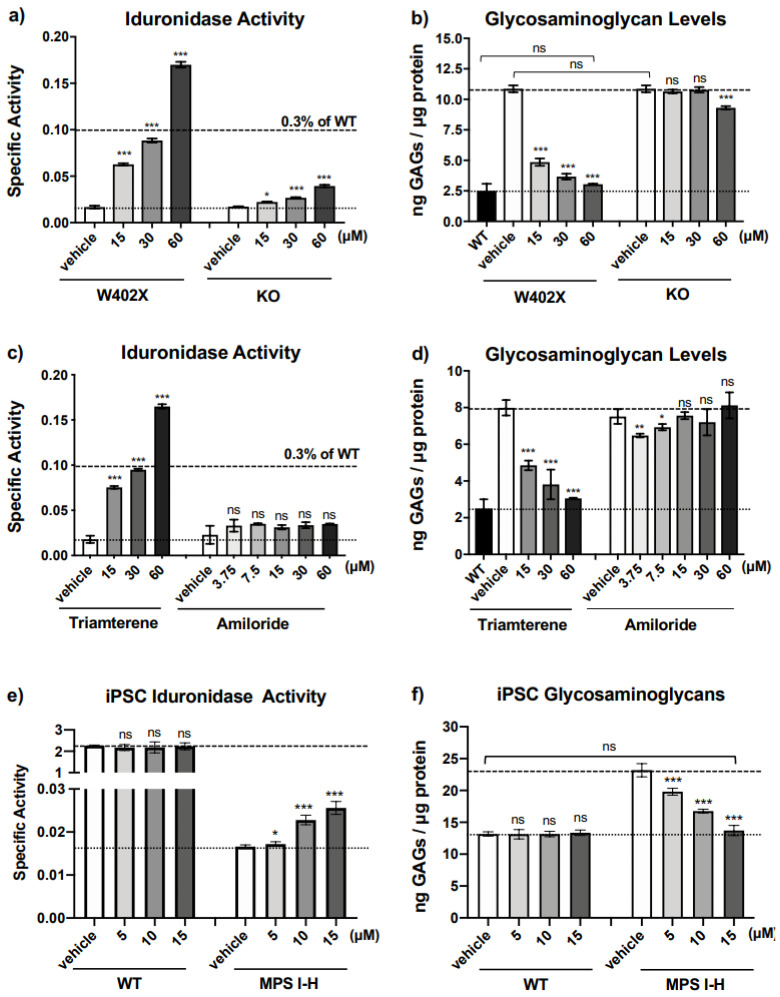
Triamterene restores α-L-iduronidase activity through a PTC-dependent manner that is unrelated to ENaC function. Cell lysates from *IDUA*-W402X and *IDUA* knock-out (KO) MEFs cultured in the presence of triamterene for 48 h were used to measure: (**a**) α-L-iduronidase specific activity, and (**b**) sulfated GAG levels [20]. Cell lysates from *IDUA*-W402X MEFs treated for 48 h with the ENaC inhibitor, amiloride, were also used to measure: (**c**) α-L-iduronidase specific activity, and (**d**) sulfated GAG levels. (**e**) The α-L-iduronidase specific activity, and (**f**) sulfated GAG levels were determined in human wild-type (WT) or MPS I-H (Q70X/W402X *IDUA* alleles) iPSCs ± triamterene treatment. α-L-iduronidase specific activity units in panels a, b, and e = nanomoles of substrate released per μg of protein per hour. Each column represents the mean ± SD of two independent experiments (n = 8–12). In panels (**a**,**c**), the upper dashed line represents 0.3% of WT α-L-iduronidase activity and the lower dashed line indicates the level of activity in untreated mutant cells. In panel e, the upper dashed line indicates the iduronidase activity in WT iPSCs while the lower dashed line indicates the iduronidase activity in the mutant control iPSCs. In panels (**b**,**d**,**f**), the upper dashed line indicates the GAG level in the control mutant MEFs, while the lower dashed line indicates the GAG level in WT MEFs. The *p*-values were calculated using a one-way ANOVA with Dunnett’s multiple comparisons tests where values for treated cells were compared to the vehicle control for each cohort or the WT control as indicated. *** *p* < 0.0001; ** *p* < 0.001; * *p* < 0.05; ns indicates *p* > 0.05.

**Figure 5 ijms-24-04521-f005:**
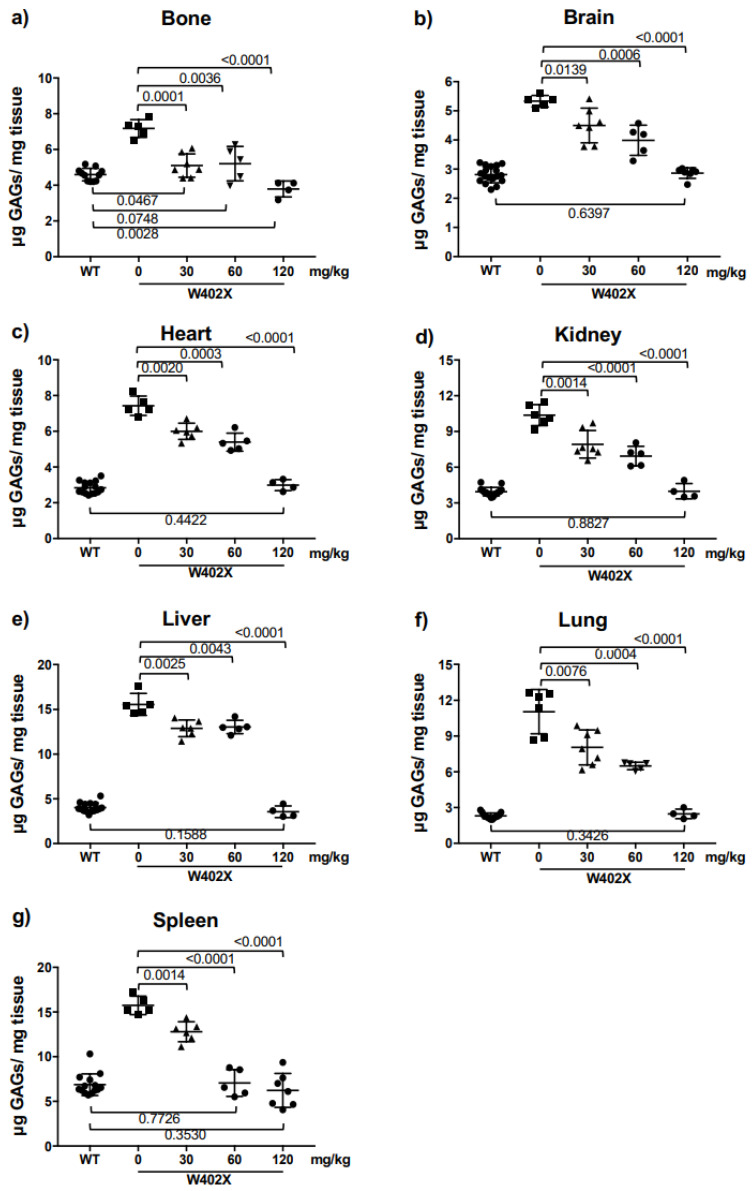
Administration of triamterene to *IDUA*-W402X mice significantly reduces GAG accumulation. Eight-week-old *IDUA*-W402X mice were administered triamterene once daily via oral gavage for a total of 2-weeks (14 doses). Mouse tissues were collected upon sacrifice, 24 h after the final dosing. Sulfated GAG levels in defatted, dried tissue homogenates were quantified using a GAG dye-binding assay [19]. GAG levels were quantified in the following mouse tissues: (**a**) bone, (**b**) brain, (**c**) heart, (**d**) kidney, (**e**) liver, (**f**) lung, and (**g**) spleen. Each point represents the average GAG value obtained from individual mouse tissue, with 5–12 mice included in each cohort. *p* values were calculated using a one-way ANOVA with Tukey’s multiple comparisons tests; the cohorts compared are indicated by the brackets. Unless otherwise indicated, *p* < 0.0001 when comparing WT and mutant cohorts.

**Figure 6 ijms-24-04521-f006:**
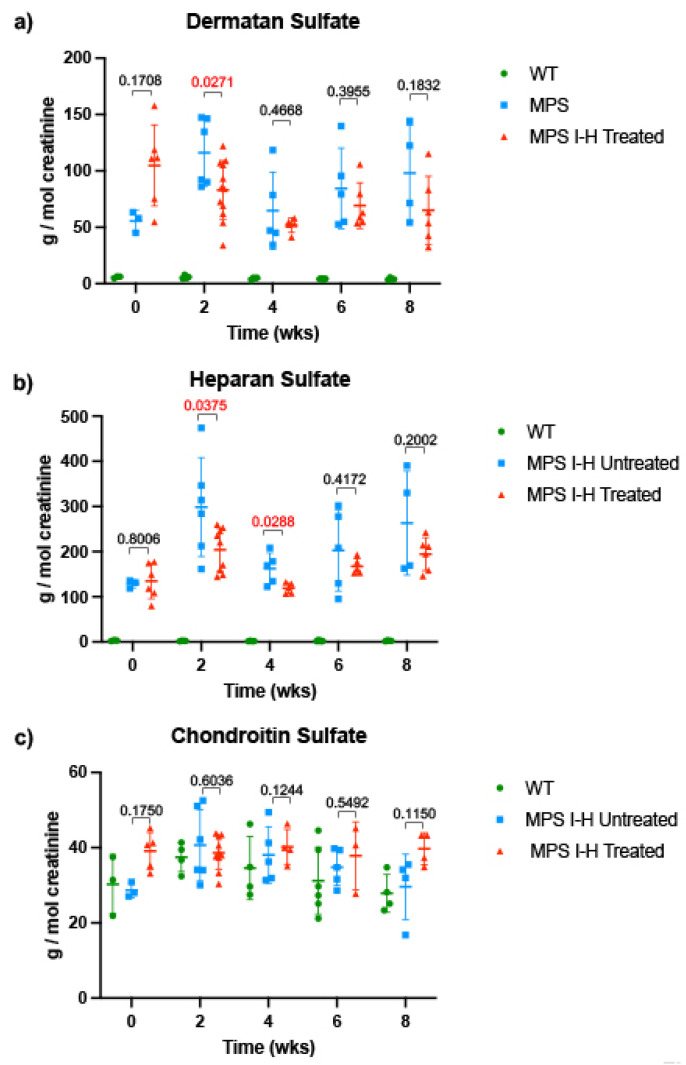
Administration of triamterene to *IDUA*-W402X mice significantly reduces urine heparan sulfate and dermatan sulfate levels. 8-week-old *IDUA*-W402X mice were administered triamterene once daily via oral gavage for a total of 8 weeks (56 doses). Urine was collected from mice 24 h before dosing began and after every 2-weeks of dosing. The (**a**) dermatan sulfate, (**b**) heparan sulfate, and (**c**) chondroitin sulfate levels were quantified using mass spectrometry and normalized to urine creatinine levels. The data is represented by the mean ± SD. 6–12 different mouse urine samples were included in each cohort. *p* values were calculated using a two-way ANOVA mixed effects comparison model with Geisser-Greenhorn correction; the cohorts compared are indicated by the brackets. Unless otherwise indicated, *p* < 0.0001 when comparing WT and mutant cohorts, with the exception of chondroitin sulfate, where no statistical differences were observed among the different cohorts (*p* > 0.05).

## Data Availability

All data for this study are shown or available upon request from the corresponding author.

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
