# Peer review of "Triamterene Functions as an Effective Nonsense Suppression Agent for MPS I-H (Hurler Syndrome)"

_ijms, 2023, doi:10.3390/ijms24054521_

Round 1

Reviewer 1 Report

This manuscript “Triamterene Functions as an effective Nonsense Suppression Agent for MPSI-H (Hurler Syndrome)” by Siddiqui and colleagues is focused on the identification of a readthrough agent for PTCs in MPS I-H. They performed a screen for PTC readthrough using a NanoLuc reporter cell line expressing in combination with 10,000 compounds. They further tested 17 of the compounds that had the greatest increase in NanoLuc with and without G418 and narrowed their “hit” list to triamterene and doxazocin. These two compounds in Idua-W402X MEFs further showed increases in Iduronidase activity and decreases CAG levels which are important markers for MPS I-H improvement. However, upon further inspection only triamterene displayed suppression of a PTC through readthrough versus doxazocin which displayed mRNA stabilization in a PTC-independent manner. Testing triamterene in the Idua-KO MEFs supported this conclusion as no markers for improvement in these cell lines occurred in the KO MEFs as expected. Additionally, the authors also showed that triamterene did not affect iduronidase activity through its ENAC blocking ability by comparing it with amiloride, another ENAC inhibitor. The final findings were in vivo in which Idua-W402X mice were treated with
triamterene for two weeks with varying doses. Significant reduction in CAGs in 6 tissues were
observed in these Idua-W402X mice after 2 weeks of treatment. Unfortunately longer treatments with triamterene did not lead to long term improvements in these mice. The authors attribute these disappointing results to increased metabolism of triamterene in the mouse liver which leads to the reduction of efficacy of this drug for PTC suppression. Interestingly, the authors
point out that triamterene may still be a viable PTC suppressor in MPS I-H patients since the
drug is already FDA approved and does not seem to be metabolized as quickly in humans. Overall, the studies were well done and produced significant findings. The comments/questions below are a few things to consider/answer to improve upon an already very solid manuscript.

General comments/questions:

1. The discussion needs more examination of other diseases originating from PTCs. Do the authors believe triamterene will be an compound that could be utilized in other diseases or would this be specific to MPS I-H? Why or why not?

2. If the disappointing long term in vivo results are due to metabolism, is there another animal model with a PTC that could be utilized or mentioned for future studies?

3. The statement in lines 156-158 “This suggests that the synergistic increases in readthrough observed with the NanoLuc reporter may be dependent on factors such as the cell type or the PTC mRNA context.” is quite interesting. Can you expand on this in the discussion? The fact that a strong effect with G418 was observed with these compounds in the NanoLuc FRT cells but no effect was observed with G418 in Idua-W402X MEFs is interesting. The reason some of these hits were interesting was their effect with G418 but that goes away in MEFs. Should the results in the FRT cells be believed? If it is the PTC mRNA context can you show the reader what the differences are between the W134X in the NanoLuc and the idua-W402X in the MEFs? Could the addition of G418 just be more toxic in MEFs which may lead to absence of synergy?

4. The authors are interested in identifying readthrough agents but could doxazocin be an important mRNA stabilizer? Is through NMD inhibition maybe? Were triamterene and doxazocin tried together at all in any of these assays?

5. Why was 60uM used for the compounds in this screen? This seems like a high dose for typical high throughput screens.

6. It might be useful to show Idua mRNA levels of some of the tissues for the 2 week treated animals. The amount of CAGs is nice but showing an increase in Idua mRNA would further support triamterene’s ability for readthrough.

Author Response

We thank the reviewers for their thoughtful reviews and constructive comments. They identified several gaps in our manuscript, which has allowed us to make several important improvements.

Reviewer 1:

  1. The discussion needs more examination of other diseases originating from PTCs. Do the authors believe triamterene will be a compound that could be utilized in other diseases or would this be specific to MPS I-H? Why or why not? We believe that triamterene could potentially be used as a readthrough agent for other disease models. We and others have shown that context effects can have a significant impact on the efficiency of PTC readthrough. While we mainly focused on the Idua-W402X context, it is important to note that triamterene alone increased readthrough by 13-fold in the NanoLuc reporter context. The NanoLuc context is significantly different from the Idua-W402X context, suggesting that triamterene can likely induce readthrough at a variety of different contexts. I have introduced a paragraph summarizing these thoughts in the discussion.
  2. If the disappointing long term in vivo results are due to metabolism, is there another animal model with a PTC that could be utilized or mentioned for future studies? Yes, we were disappointed that we could not assess long-term in vivo results. However, we don’t feel that this negates the possibility that triamterene may be effective in MPS I-H patients since it is metabolized much less extensively in humans. An MPS I-H rat model carrying a W402X mutation has recently been reported [Gurzeler et al. (2023) https://doi.org/10.1101/2023.01.31.526456], but triamterene would need to be first assessed for it metabolism in rats.
  3. The statement in lines 156-158 “This suggests that the synergistic increases in readthrough observed with the NanoLuc reporter may be dependent on factors such as the cell type or the PTC mRNA context.” is quite interesting. Can you expand on this in the discussion? The fact that a strong effect with G418 was observed with these compounds in the NanoLuc FRT cells but no effect was observed with G418 in Idua-W402X MEFs is interesting. The reason some of these hits were interesting was their effect with G418 but that goes away in MEFs. Should the results in the FRT cells be believed? If it is the PTC mRNA context can you show the reader what the differences are between the W134X in the NanoLuc and the idua-W402X in the MEFs? Could the addition of G418 just be more toxic in MEFs which may lead to absence of synergy? We don’t yet know why we are able to achieve synergy in the NanoLuc FRT reporter cells, but not in Idua-W402X MEFs. Recent studies have shown that synergy can also be detected in other differentiated cell types, including human bronchial epithelia cells and HEK293s [Sharma et al. (2021) Nat Comm 12:4358; Lee et al. JCI 132:3154571, https://doi.org/10.1172/JCI154571]. We speculate that certain aspects of translation and/or mRNA turnover might differ between embryonic and differentiated cells that could affect whether synergy occurs. For example, we have found that the extent by which Idua-W402X mRNA abundance is reduced by NMD differs between mouse embryonic tissue and tissues from adult mice. Idua-W402X mRNA abundance is reduced to around 5% of wild-type levels in MEFs, while in tissues from adult mice, Idua-W402X mRNA abundance is reduced to 25-50% of wild-type [Keeling et al. 2013, PLOS ONE 8:e60478]. I have added this paragraph to the discussion. Also, we did not see any signs of toxicity (cell death, slowed cell growth, a decrease in protein concentration) associated with the G418 concentrations that were used with the MEF experiments, so I think that is a less likely explanation for the differences in synergy observed.
  4. The authors are interested in identifying readthrough agents but could doxazocin be an important mRNA stabilizer? Is through NMD inhibition maybe? Were triamterene and doxazocin tried together at all in any of these assays? Because doxazocin increased both WT and W402X Idua mRNA abundance at the 60mM conc., we think it is likely not specific for NMD, but rather, may affect overall mRNA stability or synthesis. We did try combination studies with the iduronidase assay in the MEFs, but did not observe a significant increase in activity.
  5. Why was 60uM used for the compounds in this screen? This seems like a high dose for typical high throughput screens. 60uM is normally is the highest concentration that we use when performing a dose response curve. This concentration was used to maximize the number of hits obtained in the initial HTS screen.
  6. It might be useful to show Idua mRNA levels of some of the tissues for the 2 week treated animals. The amount of CAGs is nice but showing an increase in Idua mRNA would further support triamterene’s ability for readthrough. We agree that an increase in Idua mRNA abundance would lend further support for readthrough of the W402X PTC. However, due to the amount of tissue available for various assays, we were unable to perform both the GAG assays and mRNA abundance with the same tissues. With limited tissue availability, we felt that the GAG assay took priority since it directly correlates with restoration of iduronidase function.

Reviewer 2 Report

Siddiqui et al. describe an extensive study focused on search for chemical agents with the ability to suppress premature termination codon that will have the therapeutic potential for the severe type of mucopolysaccharisosis I, Hurler syndrome (MPS I-H). Since most patients suffering from MPS I-H carry a nonsense mutation, it is an excellent candidate for readthrough therapies.

The Authors have applied divergent methods including: high throughput screening of almost 10,000 compounds based on the NanoLuc reporter, preliminary in vitro analysis of the top 17 identified hits, more detailed in vitro tests on the mechanism of action of the two most promising agents (triamterene and doxazocin), and in vivo experiments to investigate the effects of triamterene on biochemical endpoints.

The manuscript is very well written and it is easy to follow the logic of subsequent experiments and I have only a couple of comment the Authors might wish to consider:

Figure 1 caption, line 112: shouldn’t it be “…high throughput screen (HTS)…”?

Figure 1 b and c: since x axes present here the data in the nominal scale I would suggest removing labels and changing axis title (e.g. into “screened compounds”)

Figure 1 b and c: please change the sign of “micro” in the G418 concentration next to the dotted line (now it is ug/ml)

Figure 6: “The data is represented by a box and whiskers plot where the upper and lower bars of each box represent the maximum and minimum values, respectively” (line 356-357). It is not explained what the whiskers represent and why for some data there is only a line instead of the box. Could you please add this information.

Author Response

We thank the reviewers for their thoughtful reviews and constructive comments. They identified several gaps in our manuscript, which has allowed us to make several important improvements.

Reviewer 2:

The manuscript is very well written and it is easy to follow the logic of subsequent experiments and I have only a couple of comment the Authors might wish to consider:

Figure 1 caption, line 112: shouldn’t it be “...high throughput screen (HTS)...”? The reviewer is correct. I have changed this in the Figure 1 legend.

Figure 1 b and c: since x axes present here the data in the nominal scale I would suggest removing labels and changing axis title (e.g. into “screened compounds”)

Figure 1 b and c: please change the sign of “micro” in the G418 concentration next to the dotted line (now it is ug/ml) I changed ug to ug in Figure 1.

Figure 6: “The data is represented by a box and whiskers plot where the upper and lower bars of each box represent the maximum and minimum values, respectively” (line 356-357). It is not explained what the whiskers represent and why for some data there is only a line instead of the box. Could you please add this information. Thank you for identifying these omissions. Due to formatting issues with Prism, I was unable to resolve the appearance of the box and whiskers graph. I have therefore changed the style of the graph in Figure 6 to a mean +/- SD plot where all the individual data points are shown. I also changed the description of the plots in the Figure 6 legend. I also think this presentation of the data will also be easier for the reader to interpret.